# GIS and Remote Sensing-Based Multi-Criteria Analysis for Delineation of Groundwater Potential Zones: A Case Study for Industrial Zones in Bangladesh

**Md. Mizanur Rahman** [1] , **Faisal AlThobiani** [2] , **Shamsuddin Shahid** [3] , **Salvatore Gonario Pasquale Virdis** [4] , **Mohammad Kamruzzaman** [1,*] , **Hafijur Rahaman** [5] , **Md. Abdul Momin** [6] , **Md. Belal Hossain** [7] **and Emad Ismat Ghandourah** [8]

1   Farm Machinery and Post-Harvest Technology (FMPHT) Division, Bangladesh Rice Research Institute (BRRI), Gazipur 1701, Bangladesh; mizan.brri2015@gmail.com
2   Faculty of Maritime Studies, King Abdulaziz University, Jeddah 21589, Saudi Arabia; falthobiani@kau.edu.sa
3   Department of Water and Environmental Engineering, School of Civil Engineering, Faculty of Engineering, Universiti Teknologi Malaysia, Johor Bahru 81310, Malaysia; sshahid@utm.my
4   Department of Information and Communication Technologies, School of Engineering and Technology, AIT Asian Institute of Technology, Bangkok 12120, Thailand; virdis@ait.ac.th
5   Workshop Machinery and Maintenance (WMM) Division, Bangladesh Rice Research Institute (BRRI), Gazipur 1701, Bangladesh; overahman14@gmail.com
6   Department of Agriculture, The University of Arkansas at Pine Bluff, 1200 N. University Drive, Pine Bluff, AR 71601, USA; mominm@uapb.edu
7   Irrigation and Water Management Division, Bangladesh Rice Research Institute (BRRI), Gazipur 1701, Bangladesh; belal.iwm@gmail.com
8   Department of Nuclear Engineering, Faculty of Engineering, King Abdulaziz University, Jeddah 21589, Saudi Arabia; eghandourah@kau.edu.sa
*   Correspondence: milonbrri@gmail.com; Tel.: +880-177-6422-808

**Abstract:** Groundwater is a crucial natural resource that varies in quality and quantity across Bangladesh. Increased population and urbanization place enormous demands on groundwater supplies, reducing both their quality and quantity. This research aimed to delineate the groundwater potential zone in the Gazipur district, Bangladesh, by integrating eleven thematic layers. Data and information were gathered from Landsat 8, the digital elevation model, the google earth engine, and several ancillary sources. A multi-criterion decision-making (MCDM) based analytical hierarchy process (AHP) was used in a GIS platform to estimate the groundwater potential index. The potential index values were finally classified into five sub-groups: very low, low, moderate, high, and very high to generate a groundwater water potential zone (*GWPZ*) map. The results show that groundwater potential in about 0.002% (0.026 km$^2$) of the area is very low, 3.83% (63.18 km$^2$) of the area is low, 56.2% (927.05 km$^2$) of the area is medium, 39.25% (647.46 km$^2$) of the area is high, and the rest 0.72% (11.82 km$^2$) of the area is very high. The validation of *GWPZ* maps based on the groundwater level data at 20 observation wells showed an overall accuracy of 80%. In addition, the ROC curve showed 84% accuracy of *GWPZ* maps when validated with water inventory points across the study region. Overall, this study presents an easy and practical approach for identifying groundwater potential zones, which may help improve planning and sustainable groundwater resource management.

**Keywords:** AHP; remote sensing; GIS; groundwater potential zone; Bangladesh

## 1. Introduction

Groundwater is one of the essential natural resources comprising about 34% of the total freshwater in the world [1]. It is the main water and considered less contaminated than other water sources [2]. It provides approximately half of the accessible freshwater used for daily cooking, drinking, and cleaning [3]. In Bangladesh, 97% of rural and 82%

of urban people depend on groundwater supply [4]. Around 79% of agricultural land depends on groundwater for crop production [5,6]. The amount of groundwater recharge in Bangladesh varies between 21 and 65 billion m$^3$ per year, indicating high uncertainty. Nearly 92% of all the country's water resources are external via transboundary waters. Only 2% of groundwater is recharged from renewable water resources resulting soon in high dependency and uncertainty of water [7]. Rapidly increasing population, unequal distribution of water resources, industrialization, and global warming have contributed to a significant rise in freshwater demand, resulting in water scarcity worldwide [8]. This has made the groundwater resource in a densely populated country like Bangladesh more critical [9]. Numerous areas have already become unsustainable for groundwater withdrawal [10]. The excessive withdrawal of groundwater resources increased the declining trend of groundwater levels by 0.1 to 0.5 m/yr [11]. This makes the need more urgent to determine groundwater potential zones (*GWPZ*) for proper groundwater management.

Groundwater exploration has traditionally relied on drilling, geophysical, geological, and hydrological methods, but these methods are time-consuming and expensive [12,13]. In addition, these survey methods do not use diverse factors that control the groundwater movement and occurrences [14]. Alternatively, remote sensing and the geographic information system (GIS) can solve this problem of groundwater investigation [15] by supporting the systematic, rapid, and excellent configuration to handle complex and large spatial data [16,17]. Several studies used remote sensing and GIS techniques to determine the groundwater potential zones around the globe [13,17–26]. Thematic layers such as geology, soil, drainage density, lineament density, slope, land use land cover (LULC), and rainfall were mostly used for *GWPZ* mapping in those studies.

Researchers have adopted different techniques for *GWPZ* mapping from thematic layers, such as frequency ratio [12,18,27], logistic regression [27,28]; the weight of evidence [12,28]; evidential belief function [28], artificial neural network [29], decision tree [30], random forest [31] etc. Most of these techniques are performed based on bivariate and multivariate analytical methods with restrictions in making assumptions before inspection and sensitivity determination of results [21,32]. In this context, the multi-criteria decision making (MCDM) based analytical hierarchy process (AHP) is considered as simple, rapid, precise, cost-effective, and reliable for determining *GWPZ* [33,34]. Many studies proved that MCDM is a powerful tool for decision making and can provide results with consistency, integrity, clarity, and correctness in judgments [35–39]. Many studies believe that using AHP with GIS is a useful and efficient strategy for geo-spatial data management [13,21,22].

AHP is used in this study to determine the groundwater potential zones in the Gazipur district, Bangladesh's industrial zone. Although several studies have been conducted across Bangladesh using remote sensing and GIS techniques to delineate the groundwater potential map [40–46], no study has been conducted in this industrial zone where sustainable groundwater resources management is essential for the industrial development and economy of the country. The specific objectives of this study were to (i) produce maps of different groundwater influencing factors; (ii) estimate the normalized weight value of each map based on its influence on groundwater potential; (iii) delineate groundwater potential zones and validate their accuracy. The approach proposed in this study might be found beneficial for government representatives, policymakers, and consumers in managing water resources and their applications.

## 2. Materials and Methods

### 2.1. Study Area

Gazipur district is located in the north of Dhaka city, the capital of Bangladesh (Figure 1). The study area occupies 1806.36 km$^2$ with a riverine area of 17.53 km$^2$ and forest cover of 273.42 km$^2$. The district's total population is more than 3.4 million with a density of 1884/km$^2$. The population is increasing at an annual growth rate of 5.21% [47]. The district is divided into five sub-districts called Upazila, namely Gazipur Sadar, Kaliganj, Sreepur, Kapadia, and Kalikoir. The yearly average maximum and minimum temperatures are

36 °C and 12.7 °C, respectively, with annual rainfall of 2376 mm [47]. The study area is surrounded by several rivers, namely Old Brahmaputra, Shitalakshya, Turag, Bangshi, Balu, and Banar. Gazipur is the largest industrial area in Bangladesh. Garment production, the Bangladesh Ordnance factory, the Aluminum factory, and many other factories occupy a large portion of the district, as well as a security printing press, a textile mill, a ceramics factory, packaging production, and others. Large numbers of the labor force (55%) are involved in these sectors to sustain their livelihood [48]. Table 1 represents the industrial scenario of the Gazipur district in Bangladesh.

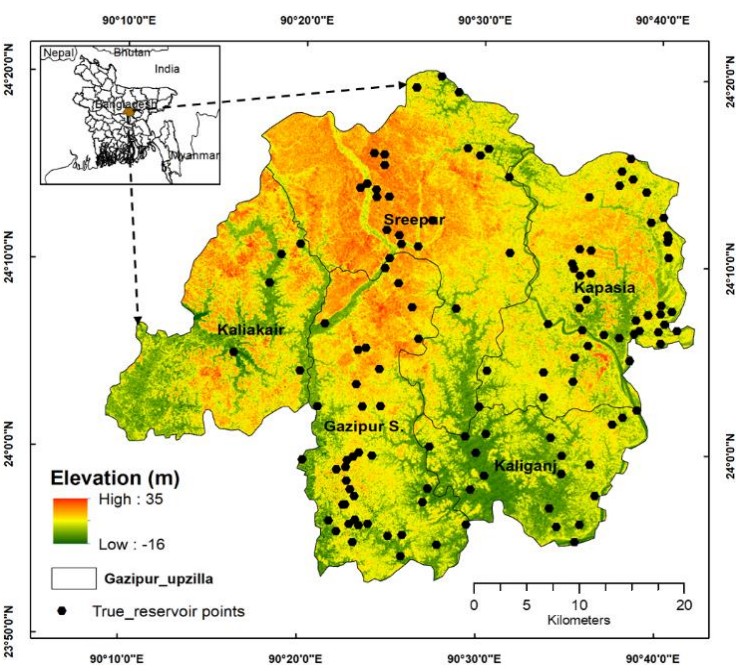

**Figure 1.** The geographic and topographic feature of the study area.

**Table 1.** Available industries in Gazipur [47].

| Upazila | Garments | Textile | Match Factory | Rice Mill | Steel and Engineering | Aluminum Factory | Jute Mill | Others |
|---|---|---|---|---|---|---|---|---|
| Gazipur Sadar | 822 | 73 | 2 | 227 | 27 | 9 | 0 | 203 |
| Kalikair | 51 | 35 | 0 | 45 | 2 | 1 | 2 | 17 |
| Kaligang | 0 | 0 | 0 | 62 | 0 | 0 | 1 | 4 |
| Kapasia | 0 | 0 | 0 | 14 | 0 | 0 | 0 | 3 |
| Sreepur | 25 | 19 | 0 | 42 | 0 | 2 | 0 | 85 |
| **Total** | **898** | **127** | **2** | **390** | **29** | **12** | **3** | **312** |

### 2.2. Thematic Layers Selection

The *GWPZ* map was created in the ArcGIS environment using remote sensing and existing datasets. Eleven thematic layers were used to delineate the *GWPZ* in this study, including geology, slope, lineament density, drainage density, rainfall, land use and land cover (LULC), soil type, soil depth, topographic wetness index (*TWI*), plane curvature, and profile curvature. The groundwater potential of a region depends on surface hydrological and sub-surface geological conditions. Higher rainfall and abundant surface water bodies (drainage density) enhance groundwater recharge potential. The topography (*TWI*, plane curvature, profile curvature, etc.) determines water accumulation on the land surface, while soil type and depth determines its percolation to the subsurface. Vegetated land and lineaments also help infiltrate surface water to subsurface more. Geology plays a

major role in accumulating and transmitting sub-surface water. Therefore, these factors were considered to determine *GWPZ* in the study area. Generally, a flat terrain covered by favorable LULC and soil with favorable underlying geology has higher groundwater potential. The potential increases when rainfall in the area is more and drainage density is less.

To prepare the geospatial datasets for this study, the required datasets were gathered from various government organizations and websites. The United States Geological Survey (USGS) earth explorer website [49] was used to collect satellite data. ASTER 30 m DEM data was used to extract the slope, profile curvature, plane curvature, drainage density, and topographic wetness index. The LULC and lineament density maps were prepared using Google Earth Engine and the line density tool in ArcGIS. The details of each factor are described below:

### 2.2.1. Slope

The slope is an important component that determines the soil's ability to absorb rainwater [50]. A higher slope value indicates less infiltration of precipitation into the topsoil to restore the groundwater aquifer. In contrast, a lower slope increases the probability of groundwater availability [51]. The slope map was created from the raster DEM data at a resolution of 30 m [52]. It should be noted that ASTER DEM is sensitive to tree canopy cover. However, it shows significant positive bias only in areas covered with dense and tall trees. The forest cover in the study area is not dense, and therefore, any bias in elevation due to the forest was ignored. Raster DEM data was processed to create the slope map of the study area using the slope tool in ArcGIS. As a result, the slope map was resized in 30 m$^2$ spatial resolution. The slope map was reclassified into five sub-classes using the natural breaks (Jenks) classification method, as shown in Figure 2. The Jenks method considers the variability of data within a group and inter-group to optimally divide the datasets into sub-groups.

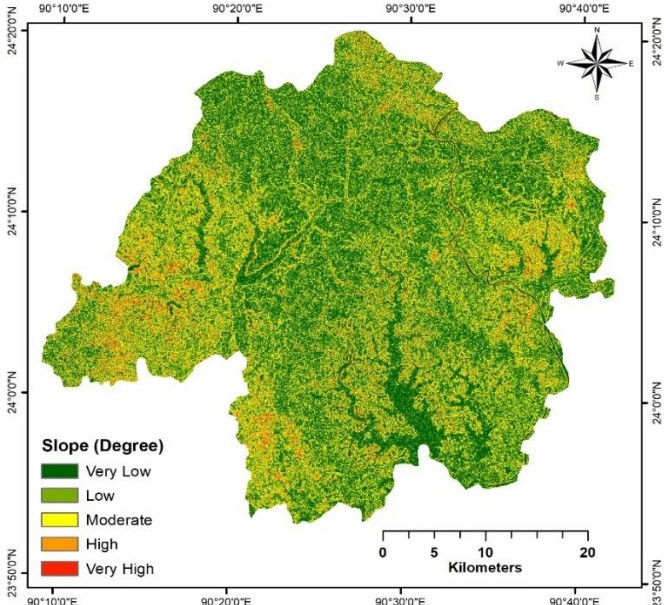

**Figure 2.** Slope map of the study area.

### 2.2.2. Drainage Density

Drainage density deals with surface runoff and permeability and is used to describe the physical factors of a drainage basin. The drainage basin is the area where rainfall accumulates and drains off into an outlet like a river or other body. The length of the steam channel is the continual form of surface water [53]. Drainage density plays an important role in delineating groundwater potentiality and contamination [54]. The higher drainage

density indicates a high runoff with low infiltration, whereas a lower drainage density indicates least runoff with high infiltration [55]. It is calculated as the total length of the steam channel in a drainage basin divided by the total area [12,52] using the following formula.

$$Dd = \frac{\sum_i^n L}{A} \qquad (1)$$

where $D_d$ is the drainage density (km/km$^2$), $L$ indicates the total length of steam (km), and $A$ is the total area (km$^2$) of the drainage basin. Finally, the drainage density map was resized to 30 m$^2$ and classified into 5 sub-classes using the natural breaks classification method (Jenks), as shown in Figure 3.

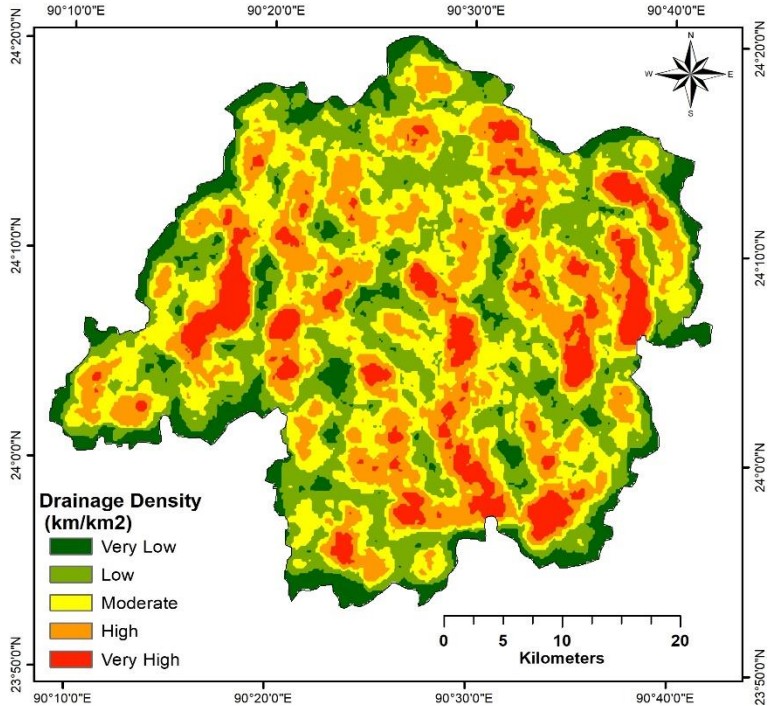

**Figure 3.** Drainage density map of the study area.

### 2.2.3. Lineament Density

Lineament is generally a representation of geologic or geomorphologic activities that produces discontinuity on the earth's surface [56]. Geological features like shear zone, dikes, faults, fractures, veins, and bedding planes help form lineament [57]. Lineament provides information on subsurface structures that regulate surface water flow, which aids in groundwater storage [58]. In this study, multispectral satellite imagery like Landsat 8 (OLI/TIR) was used to detect and extract the lineaments [57], as shown in Figure 4. The lineament is calculated as the total length of lineament in a unit area, using the below equation [22,51].

$$Ld = \frac{\sum_1^n Li}{A} \qquad (2)$$

where $Li$ indicates the length of lineament and $A$ is the unit area. The lineaments' orientation was analyzed using the rose diagram (Figure 5). The rose diagram is a bidirectional frequency diagram used to delineate the possible direction of lineaments among north, east, south, and west. Based on the length of lineaments, the rose diagram illustrates the minor and major lengths in all provable directions [59].

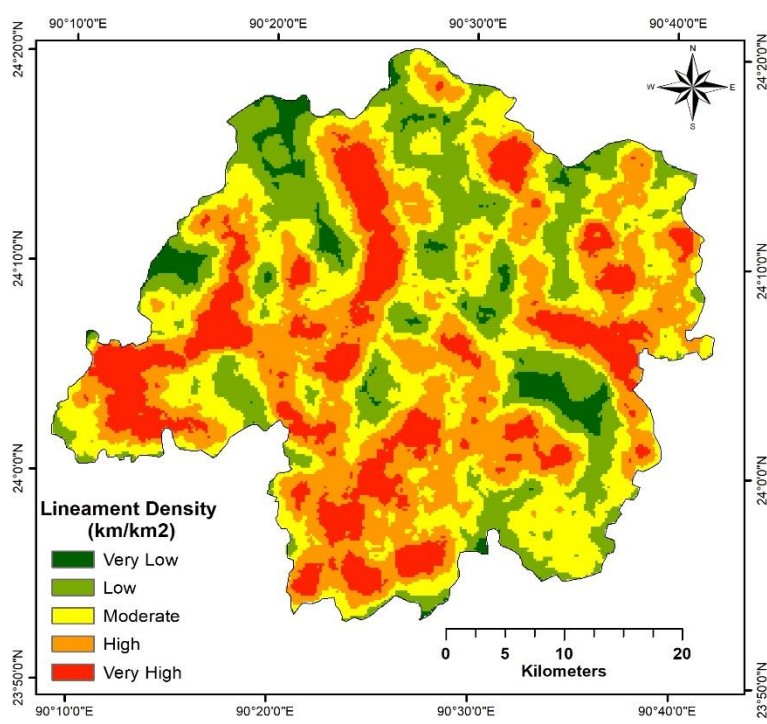

**Figure 4.** The lineament density map of the study area.

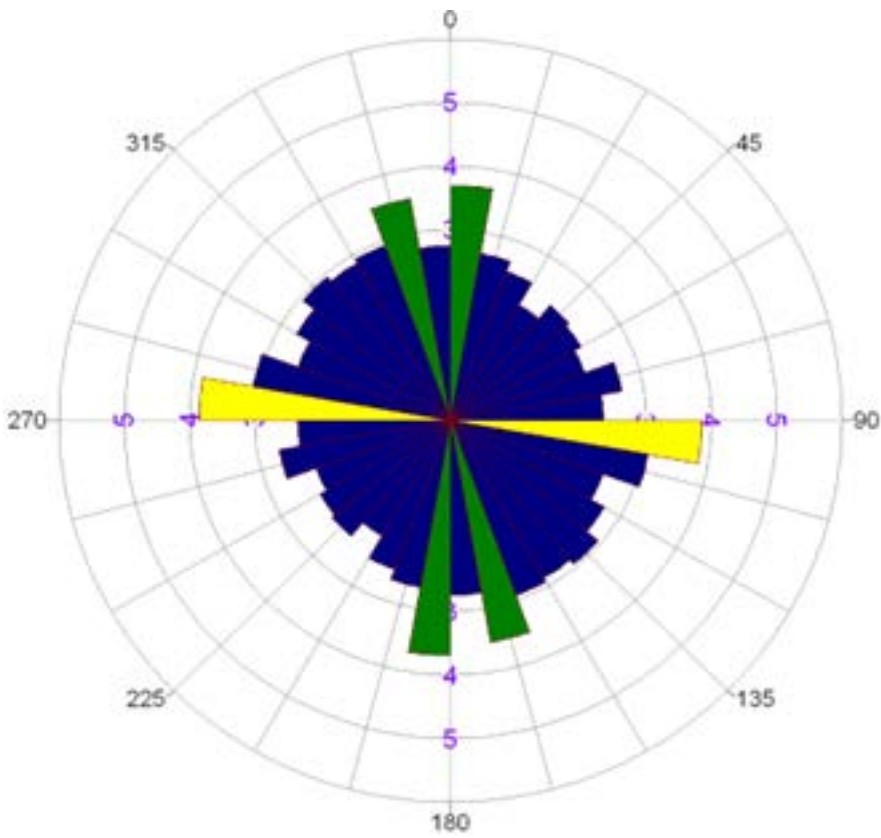

**Figure 5.** The orientation of lineament using a rose diagram.

### 2.2.4. Land Use Land Cover (LULC)

LULC is the most influencing element of groundwater incidence and development [60]. The LULC controls soil infiltration and surface runoff [61]. In this study, the google earth Engine (GEE) platform was used to generate the LULC map. GEE is a powerful cloud-based

platform for processing satellite imagery on a wide scale where no previous knowledge about large scale cloud computation is needed [62]. GEE code editor scripts were used to acquire 30 m resolution Landsat 8 OLI surface reflectance data for the target year 2021 starting from July 2020 to July 2021. A pixel-based supervised classification was used for LULC classification. The ground truth samples were collected with the help of google earth imagery to generate six classes, forest, water, agriculture, vegetation, buildup area, and barren land as shown in Figure 6.

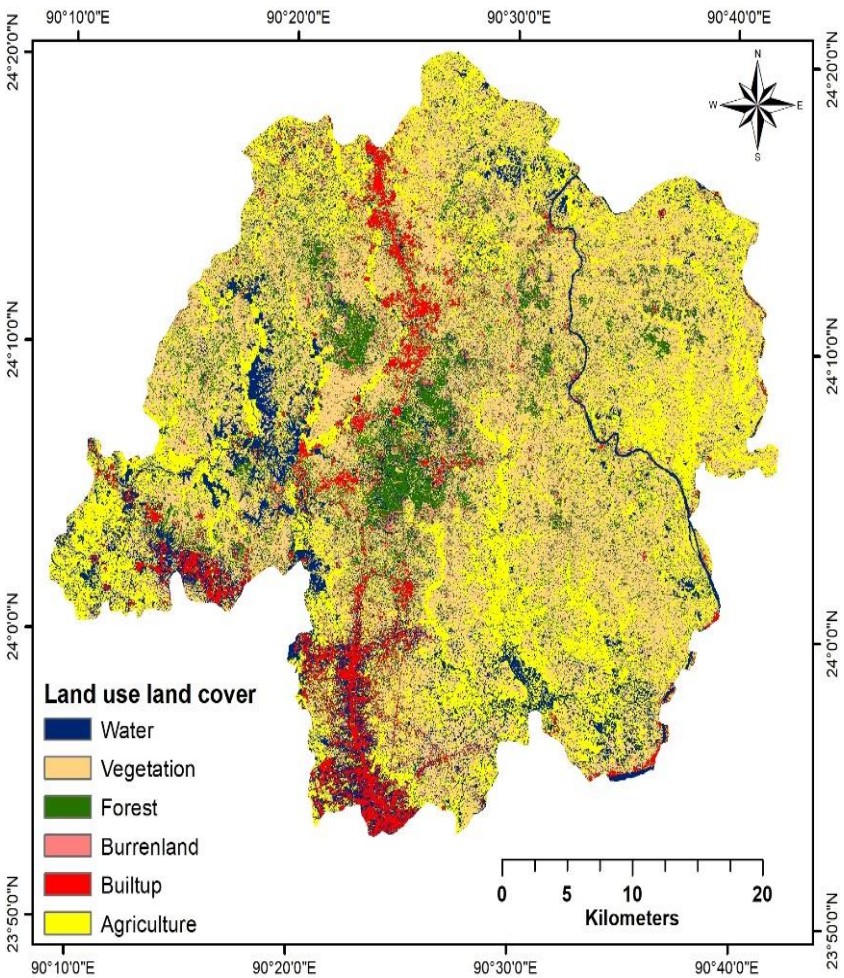

**Figure 6.** Land use land cover map of the study area.

### 2.2.5. Geology

Geological structure plays a major role in the distribution and occurrence of groundwater in any terrain [24]. Geology controls surface water penetration into the deeper ground layer [63]. The geological data was obtained from Bangladesh's digital geological and geographic data and was published by the United States geological survey's world energy project from 1997 to 2000 with a scale of 1:100,000. Then, the research region was clipped and processed in the ArcGIS environment. The dataset was resized in 30 $m^2$ spatial resolution and classified into sub-categories (Figure 7).

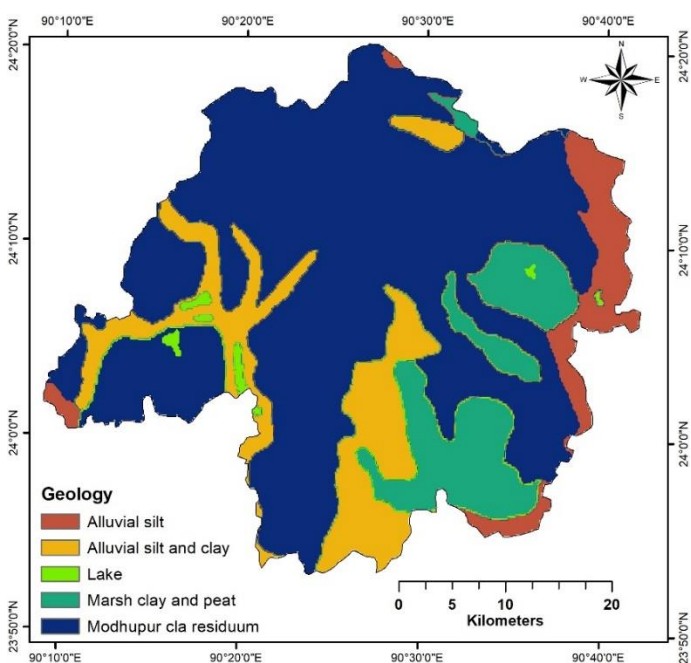

**Figure 7.** Geology map of the study area.

### 2.2.6. Soil Type

The soil texture and its hydraulic characteristics are essential for assessing the infiltration rate [24]. The soil type, soil permeability, soil moisture content, thickness, and infiltration rate are directly related to the rainfall-runoff [64]. Soil data was collected from Bangladesh Agricultural Research Council (BARC) website [65] at the national level with a scale of 1:250,000. The data was clipped using the study area boundary and then processed in an ArcGIS environment. Finally, the raster dataset was resized in 30 m spatial resolution and reclassified into three sub-categories (Figure 8).

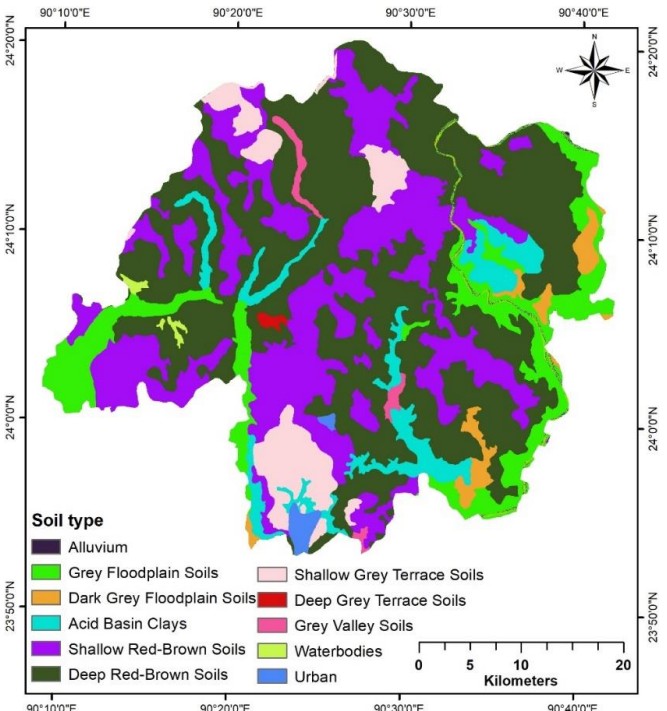

**Figure 8.** Soil type map of the study area.



### 2.2.7. Soil Depth

Soil depth is an essential element in many disciplines of earth. It has a significant function in hydrological and geological processes [66]. Soil depth governs the response of surface runoff [67], the residence of water and travelling time for distribution [68], plant-available water, storage, and sourcing [69]. A higher soil depth indicates a higher probability of groundwater potential [26]. Soil depth data was collected from BARC at a scale of 1:250,000 and processed in an ArcGIS environment. Finally, soil depth was classified into three sub-classes: shallow, medium, and deep, as shown in Figure 9.

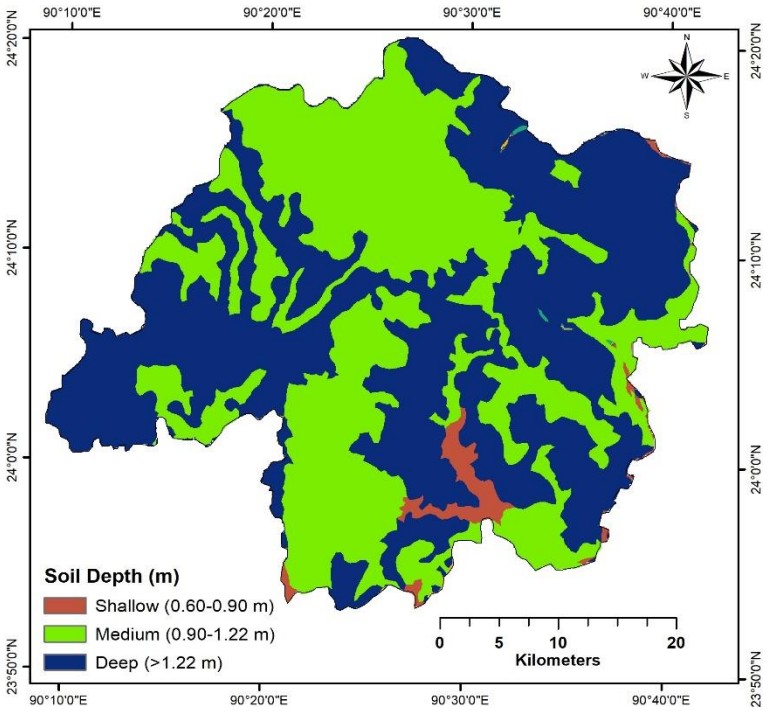

**Figure 9.** Soil depth map of the study area.

### 2.2.8. Rainfall

Rainfall is one of the important factors in determining the groundwater availability of a region [46]. The rainfall plays an important function in the hydrological cycle and controlling groundwater recharge [51]. Precipitation products derived from global model reanalysis and satellite observations may be a viable source of rainfall data in Bangladesh. According to rainfall detection metrics, ERA5 outperformed Bangladesh [70]. Kamruzzaman et al. [71] recently used ERA5 data sets to evaluate the CMIP6 global climate models in reconstructing Bangladesh's rainfall climatology. As a result, the ERA5 reanalysis dataset was used as a proxy for observation in the current study due to the unavailability of weather stations in the study area.

The ERA5 rainfall data with a spatial grid size of 30 km$^2$ was collected from the ECMWF web portal [72]. The rainfall data were entered as point data in ArcGIS, and then the rainfall map was prepared using the Inverse Distance Weighted (IDW) technique because IDW gives satisfactory results [73]. Finally, the rainfall map was resized in 30 m$^2$ spatial resolution and reclassified into 5 sub-classes using the natural breaks classification method (Jenks), as shown in Figure 10.

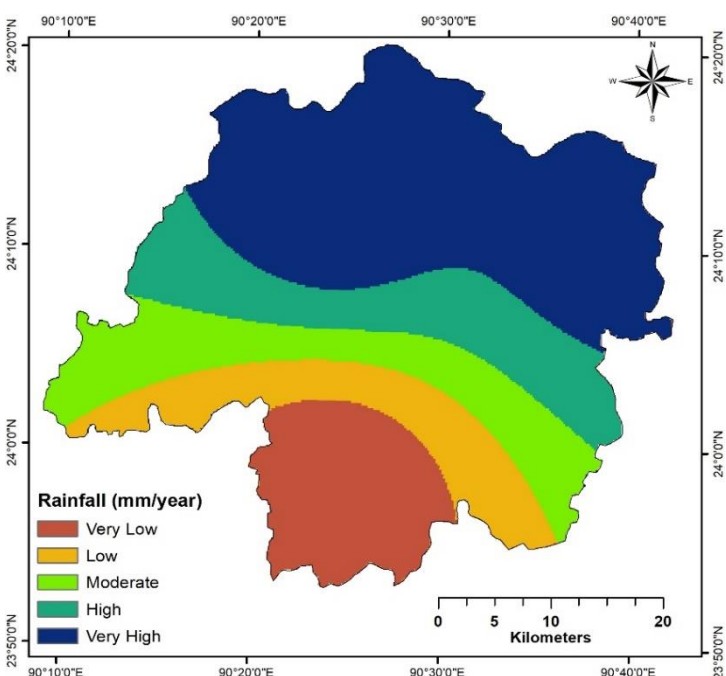

**Figure 10.** Rainfall map of the study area.

2.2.9. Topographic Wetness Index (TWI)

Figure 11 shows the topographic wetness index of the study area. *TWI*, first introduced by Arheimer et al. [74], represents steady-state wetness. It is generally used to study the hydrological process [75]. *TWI* is a unitless index and is widely used for *GWPZ* mapping [19,25,26,28,72]. A higher value of *TWI* indicates a high possibility of groundwater availability [24]. *TWI* was mapped from the slope map, generated from 30 m DEM data (Figure 11). *TWI* was generated by calculating the rate of change in a grid cell aspect compared to its neighbor using the below equation [76].

$$TWI = \ln\left(\frac{\alpha}{(tan\beta)}\right) \tag{3}$$

where the specific catchment area is $\alpha$ and the slope gradient is $tan\beta$.

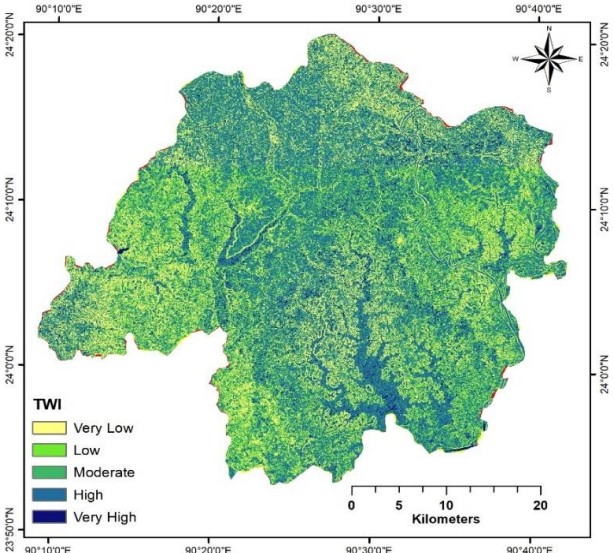

**Figure 11.** Topographic wetness index map of the study area.

### 2.2.10. The Plan and Profile Curvature

The curvature focuses on the topographic morphology and has three components namely the plan, profile, and total. The groundwater flow acceleration and deceleration are mostly dominated by the surface soil profile curvature and plan curvature [77]. The plan and profile curvature were generated using the 30 m DEM data. Finally, the plan and profile curvature raster layer were prepared in ArcGIS and reclassified into five sub-classes following ref. [78], as shown in Figure 12.

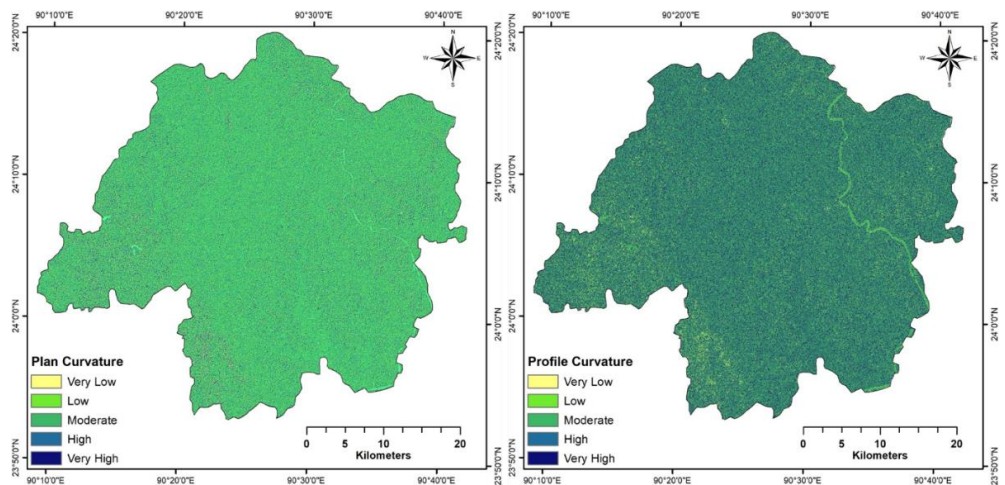

**Figure 12.** Plan curvature (**left**) and profile curvature (**right**) map of the study area.

### 2.3. Analytical Hierarchy Process (AHP) Model

Table 2 shows the scale used in the study. The paired comparisons were used to compare all of the factors in a matrix against one another. In this study, a standard scale of 1 to 9 was employed to determine the relative importance of all factors, where 1 indicates equal influence between the two factors and 9 denotes extreme influence of one factor on the other [79].

**Table 2.** Scale of comparison.

| Importance | Description |
| --- | --- |
| 1 | Equal importance |
| 2 | Equal to moderate importance |
| 3 | Moderate importance |
| 4 | Moderate to strong importance |
| 5 | Strong importance |
| 6 | Strong to very strong importance |
| 7 | Very strong importance |
| 8 | Very to extremely strong importance |
| 9 | Extreme importance |

The weight value was assigned to each factor based on the review of past studies and the influence of the water holding capacity on the groundwater potential [21,22,25,80]. The procedures below were used to determine the final weight of all of the selected theme layers [66]:

Step 1. Each column value of the pairwise matrix was added using the following formula (Table 3).

$$Lj = \sum_{i,j=1}^{n} Cij \qquad (4)$$

where $Lj$ represents the total of the values for every column of the pairwise matrix, and $Cij$ represents the number allocated to every criterion in the $i$th row and $j$th column.

**Table 3.** Pairwise comparison matrix of eleven thematic layers.

| Factors | Geology | LULC | Lineament | Drainage | Slope | Rainfall | Soil | Soil Depth | TWI | Plan Cur. | Profile Cur. |
|---|---|---|---|---|---|---|---|---|---|---|---|
| Geology | 1.000 | 1.143 | 1.333 | 1.333 | 1.333 | 1.600 | 1.600 | 1.600 | 2.000 | 2.667 | 2.667 |
| LULC | 0.875 | 1.000 | 1.167 | 1.167 | 1.167 | 1.400 | 1.400 | 1.400 | 1.750 | 2.333 | 2.333 |
| Lineament Density | 0.750 | 0.857 | 1.000 | 1.000 | 1.000 | 1.200 | 1.200 | 1.200 | 1.500 | 2.000 | 2.000 |
| Drainage Density | 0.750 | 0.857 | 1.000 | 1.000 | 1.000 | 1.200 | 1.200 | 1.200 | 1.500 | 2.000 | 2.000 |
| Slope | 0.750 | 0.857 | 1.000 | 1.000 | 1.000 | 1.200 | 1.200 | 1.200 | 1.500 | 2.000 | 2.000 |
| Rainfall | 0.625 | 0.714 | 0.833 | 0.833 | 0.833 | 1.000 | 1.000 | 1.000 | 1.250 | 1.667 | 1.667 |
| Soil | 0.625 | 0.714 | 0.833 | 0.833 | 0.833 | 1.000 | 1.000 | 1.000 | 1.250 | 1.667 | 1.667 |
| Soil depth | 0.625 | 0.714 | 0.833 | 0.833 | 0.833 | 1.000 | 1.000 | 1.000 | 1.250 | 1.667 | 1.667 |
| TWI | 0.500 | 0.571 | 0.667 | 0.667 | 0.667 | 0.800 | 0.800 | 0.800 | 1.000 | 1.333 | 1.333 |
| Plan Curvature | 0.375 | 0.429 | 0.500 | 0.500 | 0.500 | 0.600 | 0.600 | 0.600 | 0.750 | 1.000 | 1.000 |
| Profile Curvature | 0.375 | 0.429 | 0.500 | 0.500 | 0.500 | 0.600 | 0.600 | 0.600 | 0.750 | 1.000 | 1.000 |
| **Total** | **7.250** | **8.286** | **9.667** | **9.667** | **9.667** | **11.600** | **11.600** | **11.600** | **14.500** | **19.333** | **19.333** |

Step 2.  To construct a normalized pairwise matrix, every value in the matrix was divided by its column total (Table 4).

$$Xij = \frac{Cij}{Lij} \tag{5}$$

**Table 4.** Pairwise comparison matrix and normalized weight with consistency measure.

| Factors | Geology | LULC | Lineament | Drainage | Slope | Rainfall | Soil | Soil Depth | TWI | Plan Cur. | Profile Cur. | Eigen Vector | AHP Weight (%) |
|---|---|---|---|---|---|---|---|---|---|---|---|---|---|
| Geology | 0.137 | 0.138 | 0.137 | 0.137 | 0.138 | 0.137 | 0.138 | 0.138 | 0.138 | 0.138 | 0.138 | 0.138 | 14 |
| LULC | 0.121 | 0.121 | 0.121 | 0.121 | 0.121 | 0.121 | 0.121 | 0.121 | 0.121 | 0.121 | 0.121 | 0.121 | 12 |
| Lineament | 0.103 | 0.103 | 0.103 | 0.103 | 0.103 | 0.103 | 0.103 | 0.103 | 0.103 | 0.103 | 0.103 | 0.103 | 10 |
| Drainage | 0.103 | 0.103 | 0.103 | 0.103 | 0.103 | 0.103 | 0.103 | 0.103 | 0.103 | 0.103 | 0.103 | 0.103 | 10 |
| Slope | 0.103 | 0.103 | 0.103 | 0.103 | 0.103 | 0.103 | 0.103 | 0.103 | 0.103 | 0.103 | 0.103 | 0.103 | 10 |
| Rainfall | 0.086 | 0.086 | 0.086 | 0.086 | 0.086 | 0.086 | 0.086 | 0.086 | 0.086 | 0.086 | 0.086 | 0.086 | 9 |
| Soil | 0.086 | 0.086 | 0.086 | 0.086 | 0.086 | 0.086 | 0.086 | 0.086 | 0.086 | 0.086 | 0.086 | 0.086 | 9 |
| Soil depth | 0.086 | 0.086 | 0.086 | 0.086 | 0.086 | 0.086 | 0.086 | 0.086 | 0.086 | 0.086 | 0.086 | 0.086 | 9 |
| TWI | 0.069 | 0.069 | 0.068 | 0.068 | 0.069 | 0.069 | 0.069 | 0.068 | 0.069 | 0.068 | 0.068 | 0.068 | 7 |
| Plan Cur. | 0.052 | 0.052 | 0.052 | 0.052 | 0.052 | 0.052 | 0.052 | 0.052 | 0.052 | 0.052 | 0.052 | 0.052 | 5 |
| Profile Cur. | 0.052 | 0.052 | 0.052 | 0.052 | 0.052 | 0.052 | 0.052 | 0.052 | 0.052 | 0.052 | 0.052 | 0.052 | 5 |
| **Sum** | **1.00** | **1.00** | **1.00** | **1.00** | **1.00** | **1.00** | **1.00** | **1.00** | **1.00** | **1.00** | **1.00** | **1.00** | **100** |

In the normalized pairwise comparison, *Xij* signifies the value at the *i*th row and *j*th column.

Step 3. The total row of the matrix was divided by the total number of criteria using the below equation [75,76]:

$$Wi = \frac{\sum Xij}{N} \tag{6}$$

where *Wi* is the standard weight and *N* is the total criteria number.

Step 4. The consistency vector ($\lambda$) was determined by calculating the pairwise comparison matrix and the normalized pairwise matrix of the selected factors by the following formula [13,77]:

$$\lambda = \sum(Cij * Xij) \tag{7}$$

where $\lambda$ indicates the consistency vector.

Step 5. The following formula is used to calculate the consistency index (*CI*):

$$CI = \frac{\lambda max - n}{n - 1} \quad (8)$$

where *CI* indicates the consistency index and *n* is the total number of thematic layers used.
Step 6. The calculation of the consistency ratio (*CR*) was performed using this equation:

$$CR = \frac{CI}{RI} \quad (9)$$

where *RI* is the randomized index [81], as seen in Table 5 and *n* is the number of thematic layers. The *CR* < 0.1 is acceptable, while *CR* > 0.1 indicates the need to revise the pairwise comparison judgement to verify the cause of inconsistency. We obtained a *CI* value of 0.00 and a *CR* value of 0.00, which indicates that the assigned weight of the factors is perfectly consistent [82].

**Table 5.** Random index used in this study.

| *n* | 1 | 2 | 3 | 4 | 5 | 6 | 7 | 8 | 9 | 10 | 11 |
|-----|------|------|------|------|------|------|------|------|------|------|------|
| *RI* | 0.00 | 0.00 | 0.52 | 0.89 | 1.11 | 1.12 | 1.35 | 1.40 | 1.45 | 1.49 | 1.52 |

*2.4. Delineation of Groundwater Potential Zone (GWPZ)*

The weighted linear approach was applied to delineate the groundwater potential zone. The factor's weight was multiplied by the weight of feature classes of each factor. The product of all the attributes was added to obtain the groundwater potential zone (*GWPZ*) using the following formula [13,24,78]:

$$GWPZ = \sum_{i=1}^{n}(Wi * Ri) \quad (10)$$

where *GWPZ* is the groundwater potential index, *Wi* indicates the normalized weight of the factor; and *Ri* represents the weight of the features in the factor. Finally, the *GWPZ* map was prepared by categorizing *GWPZ* into five classes: very low, low, moderate, high, and very high, using the Jenks natural breaks classification in the ArcGIS environment [21].

*2.5. Validation of Groundwater Potential Zone*

A total of 20 observation wells' groundwater level data was used to validate the *GWPZ* map. The locations of the wells are shown in Figure 13. The data was collected from the Bangladesh Water Development Board, Bangladesh. The water level ranges were 4.44 to 24 m over the study region. Water table data shows the groundwater level at a point and helps determine the groundwater potential [21]. The groundwater resource potential map was prepared using the groundwater level data for validation of the potential map generated in this study. The map was generated through interpolation of water table data using the inverse distance weighting (IDW) method. The groundwater resource potential map was divided into three sub-classes namely very good zone (4.44 to 8.35), good zone (8.35 to 14.10), and poor zone (14.10 to 24.00) using the natural breaks (Jenks) classification method in the ArcGIS environment. The groundwater resource potential map was overlaid on the final *GWPZ* map [13,80], and a pixel-based analysis was conducted to correlate them. Finally, the overall accuracy of the *GWPZ* map was carried out using the following formula [37,83,84]:

$$Overall\ accuracy = \frac{Number\ of\ corrected\ obsevation\ wells}{Total\ number\ of\ observation\ wells} \times 100$$

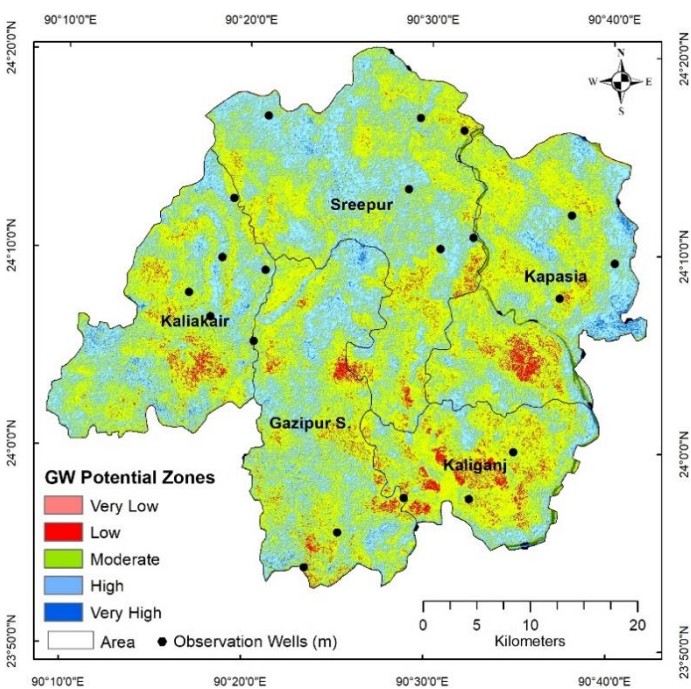

**Figure 13.** Groundwater potential zone (*GWPZ*) map of the Gazipur district.

In addition, the receiver operating characteristics (ROC) curve and area under the curve (AUC) were also utilized as performance indicators for the validation of the *GWPZ* map [21]. This study identified the number of probable groundwater sites by linking the water inventory sites to the output *GWPZ* map [85]. Based on the actual water points, the ROC curve was prepared using the ArcSDM tool in the ArcGIS environment [3,38,85–87]. The AUC value ranges from 0.5 to 1, where the value close to 1 denotes accurate measurement of the model, whereas <0.5 indicates bad performance [88].

As a final step, the TOPSIS method (order preference by similarity to the ideal solution) was used to rank the five Upazila (Gazipur Sadar, Kaliganj, Sreepur, Kapadia, and Kalikoir) of the Gazipur district in terms of groundwater potentiality. The TOPSIS was introduced by ref. [89] and is an important MCDA technique for priority based decision making [90].

## 3. Results

### 3.1. Assessment of Factors Governing Groundwater Potential Zone

#### 3.1.1. Geology

Figure 7 shows the geology map of the study area. Table 6 shows the final weight and rating values of selected factors used to determine the groundwater potential zone. Modhupur clay residuum and alluvial silt and clay comprise most of the northwest to the southwest part of the study area. The geology of the study area consists of Modhupur clay residuum covering an area of 1087 km$^2$ (65%), alluvial silt of 119 km$^2$ (7%), alluvial silt and clay of 236 km$^2$ (14%), lake of 13.14 km$^2$ (3%), and marsh clay and peat of 23.91 km$^2$ (14%). The study area mostly covers the Modhupur clay residuum. The alluvial silt was identified along the Sitalakhya river basin. Alluvial silt and clay cover the north and eastern part of the district. The marsh clay and peat were mainly observed on the eastern sides of the study region.

**Table 6.** The assigned rank and weights for the AHP method.

| Factor | Sub-Classes | Assigned Rank | Weight | AHP Rating | Total |
|---|---|---|---|---|---|
| Geology | Alluvial silt | 3 | | 0.42 | |
| | Alluvial silt and clay | 4 | | 0.56 | |
| | Lake | 5 | 0.14 | 0.70 | 2.94 |
| | Marsh clay and peat | 4 | | 0.56 | |
| | Modhupur clay residuum | 5 | | 0.70 | |
| LULC | Forest | 3 | | 0.36 | |
| | Waterbody | 5 | | 0.60 | |
| | Agriculture Land | 4 | | 0.48 | |
| | Vegetation | 3 | 0.12 | 0.36 | 2.64 |
| | Urban Area | 2 | | 0.24 | |
| | Fellow Land | 3 | | 0.36 | |
| | Settlement vegetation | 2 | | 0.24 | |
| Lineament | Very low | 2 | | 0.20 | |
| | Low | 3 | | 0.30 | |
| | Medium | 4 | 0.1 | 0.40 | 2.00 |
| | High | 5 | | 0.50 | |
| | Very high | 6 | | 0.60 | |
| Rainfall | Very low | 2 | | 0.20 | |
| | Low | 3 | | 0.30 | |
| | Medium | 4 | 0.1 | 0.40 | 2.00 |
| | High | 5 | | 0.50 | |
| | Very high | 6 | | 0.60 | |
| Soil Depth | Shallow | 5 | | 0.50 | |
| | Medium | 4 | 0.1 | 0.40 | 1.20 |
| | Deep | 3 | | 0.30 | |
| Drainage Density | Very low | 6 | | 0.54 | |
| | Low | 5 | | 0.45 | |
| | Medium | 4 | 0.09 | 0.36 | 1.80 |
| | High | 3 | | 0.27 | |
| | Very high | 2 | | 0.18 | |
| Soil | Noncalcareous Alluvium | 1 | | 0.09 | |
| | Grey Floodplain Soils | 2 | | 0.18 | |
| | Dark Grey Floodplain | 2 | | 0.18 | |
| | Acid Basin Clays | 5 | | 0.45 | |
| | Shallow Red-Brown Terrace Soils | 5 | | 0.45 | |
| | Deep Red-Brown Terrace Soils | 5 | 0.09 | 0.45 | 3.33 |
| | Shallow Grey Terrace Soils | 3 | | 0.27 | |
| | Deep Grey Terrace Soils | 3 | | 0.27 | |
| | Waterbodies | 5 | | 0.45 | |
| | Urban | 2 | | 0.18 | |
| | Grey Valley Soils | 4 | | 0.36 | |
| Slope | Very low | 6 | | 0.54 | |
| | Low | 5 | | 0.45 | |
| | Medium | 4 | 0.09 | 0.36 | 1.80 |
| | High | 3 | | 0.27 | |
| | Very high | 2 | | 0.18 | |
| *TWI* | Very low | 2 | | 0.14 | |
| | Low | 3 | | 0.21 | |
| | Medium | 4 | 0.07 | 0.28 | 1.40 |
| | High | 5 | | 0.35 | |
| | Very high | 6 | | 0.42 | |

**Table 6.** *Cont.*

| Factor | Sub-Classes | Assigned Rank | Weight | AHP Rating | Total |
|---|---|---|---|---|---|
| Plane curvature | Very low | 2 | | 0.10 | |
| | Low | 3 | | 0.15 | |
| | Medium | 4 | 0.05 | 0.20 | 1.00 |
| | High | 5 | | 0.25 | |
| | Very high | 6 | | 0.30 | |
| Profile curvature | Very low | 2 | | 0.10 | |
| | Low | 3 | | 0.15 | |
| | Medium | 4 | 0.05 | 0.20 | 1.00 |
| | High | 5 | | 0.25 | |
| | Very high | 6 | | 0.30 | |

### 3.1.2. Land Use Land Cover

Figure 6 shows the land use land cover map of the study area. The major parts are under vegetation (37.49%; 692.17 km$^2$) and agriculture (30.29%; 559.24 km$^2$) followed by water (14.41%; 266 km$^2$), forest (10.82%; 199.80 km$^2$), built-up area (4.91%; 90.75 km$^2$), and barren land (2.08%; 38.48 km$^2$). Agricultural practices enhance infiltration by enabling water to seep via pore pores in the soil, which retain water in the roots and loosen the rock and soil. Because of the lake's permeable surface and higher runoff, the infiltration rate was reduced in the built-up area and barren land. However, areas with vegetation, water bodies, and forests had a high groundwater potential, whereas built-up and barren areas had a low one [91].

### 3.1.3. Slope

Figure 2 shows the slope map of the study area. The slope map is categorized into five sub-classes very low (0–1.35), low (1.35–2.42), medium (2.42–3.77), high (3.77–5.65), and very high (5.65–22.87), respectively. The very low (38%; 636.83 km$^2$) and low (33.56; 561.35 km$^2$) and gentle slope over the study area are assigned a high score of 5 with a high potential zone for groundwater. The northwestern part mainly showed a higher slope due to high elevation. Slopes (2.42–3.77°) were considered moderate for groundwater potential, whereas slopes (3.77–5.65°) and (5.65–22.87°) were considered poor and very poor, respectively. Very poor groundwater potential areas cover around 20.39 km$^2$ (1.22%) over the study area.

### 3.1.4. Lineament Density

Figure 4 shows the lineament density map of the study area. Analysis of the rose diagram revealed that the maximum dominated lineament direction was E90–100° and S270–280°. The lineament trends were predominately in NNW-SSE and NNE-SSW. The lineament map was categorized in five classes namely very low (0–0.43 km/km$^2$) cover 62.51 km$^2$ (3.72%), low (0.44–0.72 km/km$^2$) cover 350.78 km$^2$ (20.88%), medium (0.73–0.99 km/km$^2$) cover 485.64 km$^2$ (30.10%), high (1.0–1.3 km/km$^2$) cover 505.56 km$^2$ (30.10%), and very high (1.4–2.0 km/km$^2$) cover 275.13 km$^2$ (16.38%). Medium and low lineament density dominate the study area. Groundwater development is more likely to occur in areas with a high lineament density because of the availability of recharge channels.

### 3.1.5. Drainage Density

Figure 3 indicates the drainage density map of the study area. The drainage density is categorized in very low (0.003–0.63 km/km$^2$) covering 184.58 km$^2$ (10.99%), low (0.63–0.94 km/km$^2$) covering 468.97 km$^2$ (27.93%), medium (0.94–1.23 km/km$^2$) covering 471.70 km$^2$ (28.09%), and high (1.23–1.57 km/km$^2$) covering 152.91 km$^2$ (9.11%). Finally, the high-rank values were applied in the very low-density area due to a greater infiltration rate [92]. Very high

drainage density was observed in the most central part, and very low drainage density was found near the administrative boundary of the district.

### 3.1.6. Rainfall

Figure 10 illustrates the rainfall map of the study area. The rainfall map is prepared and classified in five sub-classes namely very low (2193.05–2225.62 mm/year) covering 231.64 km$^2$ (13.79%), low (2225.62–2290.77 mm/year) covering 268.85 km$^2$ (16.01%), high (2290.77–2321.53 mm/year) covering 281.69 km$^2$ (16.77%), and very high (2321.53–2346.86) covering 691.09 km$^2$ (41.15%) of the study area. The higher rank was assigned to higher rainfall zones than lower rainfall areas. Results indicated that relatively high rainfall was observed on the northern sides and low rainfall occurs in the southeastern part. High and medium rainfall areas mostly cover the east to west parts of the study area.

### 3.1.7. Soil Type

Figure 8 indicates the soil type map of the study area. Based on the soil map, the study area is largely covered by deep red-brown terrace soils and covers 774.49 km$^2$ (46.73%). Nearly 440.51 km$^2$ (26.58%) of shallow red-brown terrace soil was observed at the center of the south to the northern part of the study area. Additional dominant soil types included non-calcareous grey floodplain soils (10.69%), acid basin clays (6%), and shallow grey terrace soils (5.95%). The infiltration capacity of non-calcareous brown, grey, and dark grey soils can be improved through use of calcareous and non-calcareous soils. Groundwater has a greater capacity to infiltrate and be stored than previously thought. Deep grey terrace soils and shallow grey terrace soils located in the lower southern and upper northern sections of the state, on the other hand, are less appropriate for groundwater due to their impermeable qualities [52].

### 3.1.8. Soil Depth

Figure 9 shows the soil depth map of the study area. The soil depth was categorized in shallow 29.33 km$^2$ (0.60–0.90 m), medium 745.44 km$^2$ (0.90–1.22 m), and deep 900.32 km$^2$ (>1.22 m), respectively. Shallow soil depth areas were mainly observed in the northern and eastern portions of the region. The central part of the district mostly covered the lower soil depth areas. Medium and high soil depth areas (275.40 km$^2$ and 244.21 km$^2$) were mostly in the southwest part of the district, whereas very high areas (194.52 km$^2$) were in the western part of the study area.

### 3.1.9. Topographic Wetness Index

Figure 11 shows the topographic wetness index map of the study area. The study area's *TWI* ranged from 5.84 to 16.90. As shown in Figure 6, the *TWI* values were reclassified into five categories: very high (10.49–16.90) covering 52.52 km$^2$, high (9.30–10.49) covering 126.61 km$^2$, medium (8.15–9.30) covering 198.48 km$^2$, low (7.05–8.15) covering 690.38 km$^2$, and very low (5.84–7.05) covering 583.44 km$^2$. This considerably influences drainage flow and water accumulation at the soil surface. The study area is mostly covered by high and medium *TWI*. A lower slope area is represented by a higher *TWI* rating. As a result, *TWI* is positively connected with groundwater occurrence, showing a larger groundwater potential as *TWI* increases.

### 3.1.10. Plan and Profile Curvature

Figure 12 shows the plan curvature (left) and profile curvature (right) map of the study area. Water has a natural tendency to slow down and accumulate in convex and concave profiles. During periods of heavy rainfall, a concave slope holds more water and retains it for a longer period, which is especially beneficial. The plan curvature was divided into five classes, very high (0.178 to 2.05), high (0.05 to 0.18), medium (−0.07 to 0.053), low (−0.21 to −0.0711), and very low (−2.46 to −0.21), respectively. Similarly, profile curvature was divided into five classes namely very high (0.15 to 1.84), high (0.02 to 0.15), medium

(−0.10 to 0.03), low (−0.27 to −0.10), and very low (−2.79 to −0.27), respectively. A higher curvature profile indicates a higher groundwater water probability than a lower curvature profile.

### 3.2. Groundwater Potential Map

Figure 13 shows the assigned weightage based *GWPZ* map. During the weightage calculation, the pairwise comparison matrix was constructed by following the geographic location and study area's condition. The comparison matrix was used to estimate the relative relevance of each component. The *GWPZ* map using the AHP method was obtained using Equation (7). The *GWPZ* was divided into five sub-classes namely very low, low, medium, high, and very high, using the natural breaks (Jenks) classification method in the ArcGIS environment as shown in Table 7. The potential area of groundwater was obtained by calculating the total number of pixel values using the field calculator in ArcGIS. Table 7 shows that about 0.002% (0.028 km$^2$) of the area was classified as very low groundwater potential, 3.83% (63.18 km$^2$) as low, 56.2% (927.05 km$^2$) as medium, 39.25% (647.46 km$^2$) as high, and the rest 0.72% (11.82 km$^2$) as very high groundwater potential. The study area indicated only 0.028 km$^2$ of the area as very low groundwater potential, which may be due to the elevation (−16 to 35 m), as shown in Figure 1. However, any sporadic value can generate an individual class based on the natural break algorithm due to its different statistical properties from the rest of the data. A low potential index was found only in several small polygons in this study.

**Table 7.** The available groundwater potential zone in Gazipur district.

| Potential Level | Total Area (km$^2$) | Area (%) |
|---|---|---|
| Very Low | 0.028 | 0.002 |
| Low | 63.182 | 3.830 |
| Medium | 927.047 | 56.201 |
| High | 647.456 | 39.251 |
| Very High | 11.815 | 0.716 |
| **total** | **1649.528** | **100** |

To verify the general picture of groundwater, groundwater availability was assessed at the sub-district level in the research region. The TOPSIS analysis performance index was used for this purpose. Table 8 depicts the sub-district distributions of the groundwater potential zone across the research region. The sub-district level assessment shows that Kaliganj and Gazipur Sadar are the most vulnerable zone for groundwater availability. In contrast, Sreepur and Kapasia showed high groundwater availability.

**Table 8.** Sub-district-based distributions of groundwater potential zone (%) and performance index of TOPSIS analysis.

| Sub District | Potential Level | | | | | TOPSIS Analysis | |
|---|---|---|---|---|---|---|---|
| | Very Low | Low | Moderate | High | Very High | Performance Index | Rank |
| Gazipur | 0.00 | 3.73 | 54.14 | 41.42 | 0.70 | 0.52 | 4 |
| Sreepur | 0.00 | 2.05 | 52.27 | 45.33 | 0.34 | 0.80 | 1 |
| Kapasia | 0.00 | 3.92 | 58.15 | 36.57 | 1.36 | 0.62 | 2 |
| Kaliganj | 0.00 | 8.95 | 68.76 | 21.94 | 0.35 | 0.39 | 5 |
| Kaliakoir | 0.00 | 3.22 | 54.16 | 41.82 | 0.80 | 0.61 | 3 |

### 3.3. Validation of the Groundwater Potential Zone

The groundwater potential map was validated using the 20 observation wells' water level data. The observation wells' data was used to generate a map of water level using the IDW technique. The water level map was overlaid on the groundwater potential zone map to confirm its accuracy. During validation, the outputs demonstrated that the high and

extremely high groundwater potential zones were found in locations where groundwater levels are from 4.44 to 8.35 m. Groundwater levels ranging from 8.35 to 14.10 m indicated a moderate to high potential for groundwater. Between 14.10 and 24 m, the water level was considered to have a low to very low potential. Table 9 shows the details of observation wells' data and their correlation with the *GWPZ* map. Out of 20 observation wells, 16 were highly correlated (80%) with the *GWPZ*. The rest of the four observation wells did not match the resulting maps.

**Table 9.** Details of observation wells and pixel correlation with resulting map.

| Upazila | Well ID | Lat | Lon | Water Table (m) | Actual | AHP Model | |
| | | | | | | Class | Remarks |
| --- | --- | --- | --- | --- | --- | --- | --- |
| Gazipur Sadar | GT3330001 | 23.93 | 90.42 | 7.98 | Very Good | High | Agreed |
| Gazipur Sadar | GT3330002 | 23.96 | 90.48 | 13.01 | Good | High | Agreed |
| Gazipur Sadar | GT3330020 | 23.9 | 90.39 | 24.00 | Poor | Moderate | Not Agreed |
| Kaliakair | GT3332003 | 24.11 | 90.3 | 10.62 | Good | High | Agreed |
| Kaliakair | GT3332004 | 24.21 | 90.32 | 8.23 | Good | Moderate | Agreed |
| Kaliakair | GT3332005 | 24.16 | 90.31 | 10.25 | Good | High | Agreed |
| Kaliakair | GT3332006 | 24.09 | 90.34 | 6.51 | Very Good | High | Agreed |
| Kaliakair | GT3332007 | 24.15 | 90.35 | 5.68 | Very Good | High | Agreed |
| Kaliakair | GT3332008 | 24.13 | 90.28 | 10.03 | Good | Moderate | Agreed |
| Kaliganj | GT3334009 | 23.96 | 90.54 | 4.44 | Very Good | Moderate | Not Agreed |
| Kaliganj | GT3334010 | 24.00 | 90.58 | 7.45 | Very Good | Moderate | Not Agreed |
| Kapasia | GT3336011 | 24.2 | 90.63 | 5.11 | Very Good | High | Agreed |
| Kapasia | GT3336012 | 24.16 | 90.67 | 4.73 | Very Good | High | Agreed |
| Kapasia | GT3336013 | 24.13 | 90.62 | 4.51 | Very Good | High | Agreed |
| Sreepur | GT3386014 | 24.22 | 90.48 | 5.96 | Very Good | High | Agreed |
| Sreepur | GT3386015 | 24.18 | 90.54 | 5.99 | Very Good | Moderate | Not Agreed |
| Sreepur | GT3386017 | 24.17 | 90.51 | 9.83 | Good | High | Agreed |
| Sreepur | GT3386018 | 24.27 | 90.53 | 7.78 | Good | High | Agreed |
| Sreepur | GT3386019 | 24.28 | 90.35 | 9.52 | Good | High | Agreed |
| Kaliganj | GT6768008 | 24.28 | 90.49 | 7.27 | Very Good | High | Agreed |

The receiver operating characteristics (ROC) curve analysis technique was applied to validate the output maps based on the water inventory points across the research region. However, 127 possible groundwater sites were selected and used to predict the accuracy of the *GWPZ* map (Figure 1). Figure 14 shows the ROC curve of the *GWPZ* that has a prediction accuracy of 84%. The results of the selected model clearly showed a significant correlation with the real-world data. The prediction rate of the AHP model indicated that the proposed technique is viable in terms of the delineation of the groundwater potential zone. The findings suggest that the technique used for delineation of *GWPZ* in this study is beneficial and may be used to improve water management planning in Bangladesh.

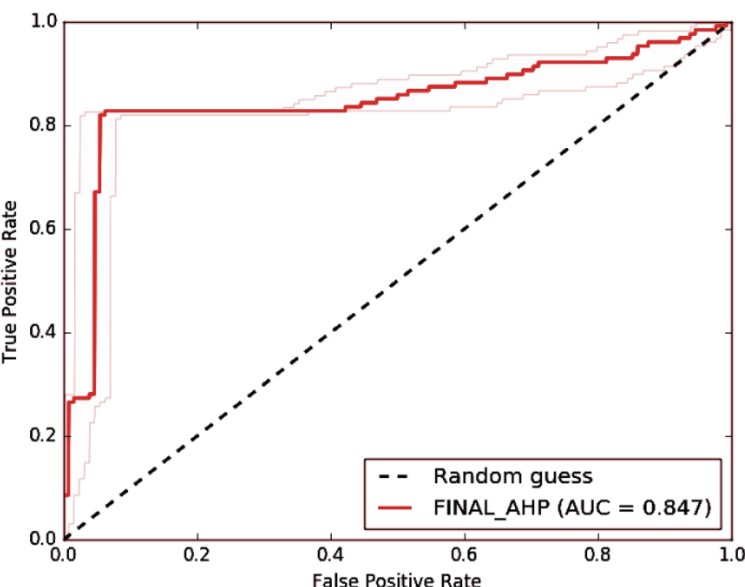

**Figure 14.** Evaluation of *GWPZ* map accuracy using AUC/ROC curve.

## 4. Discussion

This research investigated the groundwater potential of the Gazipur district of Bangladesh. The AHP method was applied to calculate the weightage values of the slope, geology, lineament density, drainage density, rainfall, LULC, type of soil, soil depth, *TWI*, plane curvature, and profile curvature. The resulted outputs indicated that selected factors have a combined effect on groundwater potential, but geology, LULC, drainage density, and lineament density were the most influential [93]. The results indicated that the Gazipur district has an overall good prospect for groundwater. The study area is covered by moderate and high groundwater potentiality of 56.2% and 39.25% respectively.

The TOPSIS analysis was incorporated here to determine the overall scenario of groundwater potential at the sub-district level. Sreepur was ranked first in terms of groundwater potential (Table 8). The combination of deep red-brown terrace soils and shallow red-brown terrace soils with slopes less than 2.42° has resulted in a region with moderate to high groundwater potential in this area. The northeast and south are dominated by deep red-brown terrace soils, whilst the center is dominated by shallow red-brown terrace soils. Deep red-brown terrace soils and shallow red-brown terrace soils are both well-drained, which has a substantial impact on groundwater potential [94]. The Sreepur indicated that 97.94% of areas belong to moderate to a very high groundwater potential zone, implying that the area is suitable for bulk groundwater exploitation.

The Kapasia and Kaliakoir were ranked second and third in groundwater potential. The moderate to higher lineament densities, deep red-brown terrace soils dominate half of these areas, and non-calcareous floodplain soils in the center part with shallow red-brown terrace soils have made the areas have a moderate groundwater potential. The non-calcareous grey floodplain soils have a good drainage capacity. Most parts of the areas belong to gentle slopes 0–1.35° [95,96] and are categorized as flatlands with no denudation. The croplands are mainly distributed in the gentle slope area. These areas are indicated as a prospect of groundwater potential.

Finally, the Gazipur and Kaliganj were ranked fourth and fifth. The presence of high drainage density, low lineament density with higher slope, low precipitation, and low topographic wetness index (*TWI*) were the principal factors for the poor groundwater potential of these areas. The settlements and barren lands also indicated poor groundwater possibility [95,97]. According to the Bangladesh Agricultural Development Corporation (BADC), Gazipur Sadar and Kaliganj suffer from groundwater depletion, which correlates with the current study's findings. This research would be helpful for groundwater planning

such as suitable site selection for industrial development, the proper place for installing the bore or dug wells, etc.

The AHP model showed an overall accuracy of 80% when validated with existing observation wells' data. On the other hand, the AHP model led to *GWPZ* maps with ROC values of 84% when validated with water inventory points across the study region. In both cases, the prediction accuracy of the models can be categorized as very good. The results indicate that the GIS and AHP-based techniques for delineating groundwater potential zones used in this study are effective and can be used to improve water management planning and development in tropical and sub-tropical regions with diverse geo-environmental settings.

## 5. Conclusions

The current study delineates the groundwater potential zones in the industrial region of Bangladesh using AHP in GIS and remote sensing platforms. The relevant studies were reviewed during the methodological design phase of the study. Eleven thematic layer geology, lineament density, general soil type, slope, drainage density, rainfall, soil depth, topographic wetness index, profile curvature, plane curvature, and land use land cover, were used to delineate the potential of groundwater. The target thematic layers were analyzed, and weights were assigned based on the findings of previous studies and their importance in defining groundwater potential. The output maps of the study region were divided into five potential zones: very high, high, moderate, low, and very low. According to the findings, about 56.2% and 39.25% of the regions had moderate to high groundwater potential which demonstrated the good prediction accuracy of *GWPZ* in the study region.

The AHP model was formulated based on the Landsat 8 OLI images, DEM of USGS, and other ancillary data sources to attain the highest prediction accuracy. However, higher resolution satellite imagery, higher precision, and the number of observation data and machine learning techniques could improve the groundwater modelling, which could not be applied in this study due to logistical barriers. Despite these obstacles, the groundwater map might benefit water resource planning and industrial growth in the studied region. Finally, the current study may be valuable for decision-makers, planners, and the government in better planning and sustainable groundwater management for multi-purpose usage.

**Author Contributions:** M.M.R. and M.K. designed the research, analyzed the data, and wrote the manuscript; S.S. and S.G.P.V. assisted in the preparation of the manuscript and subsequent revisions; H.R. and M.A.M. contributed significant intellectual content; M.K. and S.S. supervised the study and provided critical evaluations of the manuscript; M.B.H. contributed data; F.A. and E.I.G. provided funding. All authors have read and agreed to the published version of the manuscript.

**Funding:** This research received no external funding.

**Institutional Review Board Statement:** Not applicable.

**Informed Consent Statement:** Not applicable.

**Data Availability Statement:** The data used in this study are available in corresponding author. Anybody can get in on request in email to milonbrri@gmail.com.

**Conflicts of Interest:** The authors declare no conflict of interest.

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
