# Peer review of "GIS and Remote Sensing-Based Multi-Criteria Analysis for Delineation of Groundwater Potential Zones: A Case Study for Industrial Zones in Bangladesh"

_sustainability, doi:10.3390/su14116667_

Round 1

Reviewer 1 Report

This paper reports a groundwater potential zone mapping procedure with AHP which is widely used in MCDM applications. Eleven geographic and geologic factors were selected and corresponding data in the study area was collected and processed. The data processing and results validation are well presented. The paper is well structured.

I do not have expertise in justifying the correlation between the factors/layers listed in this work and hydrogeology. One thing is that rainfall is described as ‘the most important factor’. However the corresponding weight value is not the largest one.

  1. The notation of population number on line 108 is weird? And also the scale ratio on line 198.
  2. What is “upazila”?
  3. What are the definitions of a stream and a unit for Ls and A in calculation of drainage density?
  4. How does the reference [61] support ‘to detect and extract the lineaments’ mentioned on line163?
  5. ASTER DEM data is a digital surface model. The slope calculated from this one may differ from the slope of bare earth surface, especially for forest or vegetation covered regions. This happens to TWI, plan and profile curvatures as well.
  6. Over sampling may introduce uncertainties when a dataset of small scales or coarser resolutions to be interpolated to a much finer resolution one. How did this study manage this issue and control accuracies when the authors prepared some layers, for example soil type, soil depth, rainfall (30km2 to 30m?)
  7. How are the break values for some factors determined? For example sub-groups of slope, drainage density, rainfall, etc.
  8. Five classes were generated in the final groundwater potential map. Please elaborate more details justifying the break values. And what are the considerations for producing a very low class accounting for 0.028 km2?
  9. Line 575, “groundwater available zones”?

Author Response

Response to Reviewer -1 Comments

Comments and Suggestions for Authors

This paper reports a groundwater potential zone mapping procedure with AHP which is widely used in MCDM applications. Eleven geographic and geologic factors were selected and corresponding data in the study area was collected and processed. The data processing and results validation are well presented. The paper is well structured.

I do not have expertise in justifying the correlation between the factors/layers listed in this work and hydrogeology. One thing is that rainfall is described as ‘the most important factor’. However the corresponding weight value is not the largest one.

Author Response:

Thanks a lot for your valuable comments. According to your comments, we have revised the sentences as “Rainfall is one of the important factor in determining the groundwater availability of a region” (Page 11, Line:254)

Point by point answering to the reviewer comments

Point-1: The notation of population number on line 108 is weird? And also the scale ratio on line 198.

Response: Thank you very much for your valuable comments and suggestions. We changed the population number and scale ratio in the revised manuscript as per your suggestions. Please see lines 107 (page 3) and 223 (page 8).

Point-2: What is “upazila”?

Response: Thank you for raising this issue. The study area (Gazipur district) consists of five sub-districts called upazila. We elaborated this in the revised manuscript, Line 108-110 (page 3).

Point-3: What are the definitions of a stream and a unit for Ls and A in calculation of drainage density?

Response: We highly appreciated for your coment. We provided proper definitions of stream, stream length and area in our corrected manuscript. Line 165-169 (page 5)

Point-4: How does the reference [61] support ‘to detect and extract the lineaments’ mentioned on line163?

Response: Thank you for your comment. We are sorry for cited this reference in our manuscript. We already removed this reference from the revised manuscript. Line 188 (page 6)

Point-5: ASTER DEM data is a digital surface model. The slope calculated from this one may differ from the slope of bare earth surface, especially for forest or vegetation covered regions. This happens to TWI, plan and profile curvatures as well.

Response: Thank you very much for your comment. We agree that ASTER DEM is sensitive to forest cover. However, the forest cover in the study area is not dense. We added following text to justify the use of ASTER DEM:

“It should be noted that ASTER DEM is sensitive to tree canopy cover. However, it shows significant positive bias only in areas covered with dense and tall trees. The forest cover in the study area is not dense, and therefore, any bias in elevation due to the forest was ignored.” Line 152-159 (page 5)

Point-6: Over sampling may introduce uncertainties when a dataset of small scales or coarser resolutions to be interpolated to a much finer resolution one. How did this study manage this issue and control accuracies when the authors prepared some layers, for example soil type, soil depth, rainfall (30km2 to 30m?)

Response: Thank you for your comments. The data and map of soil type and soil depth were collected from the Bangladesh Agricultural Research Council (BARC) on a scale of 1:250,000. After that, above mentioned layers were processed in ArcGIS envirnment and resample in 30 m2 spatial resolution. The resolution of original data was sufficient enough and therefore, uncertainty in resampling was ignored.

Point-7: How are the break values for some factors determined? For example sub-groups of slope, drainage density, rainfall, etc.

Response: Thank you for your comments. We used Jenks natural break algorithm in ArcGIS environment for classification of data. The Jenks method considers the variability of data within a group and inter-group to divide the datasets in sub-groups optimally. We mention this in the revised manuscript. Line 157-159 (page 4) for slope, line 176-178 (page 5) for drainage density and line 267-269 (page 12) for rainfall.

Point-8: Five classes were generated in the final groundwater potential map. Please elaborate more details justifying the break values. And what are the considerations for producing a very low class accounting for 0.028 km2?

Response: Thank you for raising this issue. The final GWPZ was categorised into five classes: very low, low, moderate, high, and very high, using the natural breaks (Jenks ) classification technique in the ArcGIS environment. It is an algorithm that optimally divides the data into sub-group considering the statistical properties of the data. We mention this in the revised manuscript, where natural break classification is first described. Any sporadic value can generate an individual class based on the natural break algorithm due to its different statistical properties from the rest of the data.This also happens in our case. Low potential index was only in some small polygons. Line 521-526 (page 21).

Point-9: Line 575, “groundwater available zones”?

Response: We are sorry for using the sentence “groundwater available zones”.. We revised the sentence as “groundwater potential zone” in our revised manuscript. Line 625 page 25

Reviewer 2 Report

I have completed the reviewing of the manuscript “GIS and remote sensing-based multi-criteria analysis for delineation of groundwater potential zones: a case study for indus-3 trial zones in Bangladesh 4”. I have two questions for the authors as follows:

  • What rules are the predicting factors selected based on? The author should give some explanations?
  • What are the relations between the selected factors?
  • The AHP is a semi-subjective or subjective method frequently used in many decision-making system? The author should discuss its limitation. Or compare its performance with similar studies like:
  1. Flood Risk Assessment in Metro Systems of Mega-Cities Using a GIS-Based Modeling Approach.
  2. A comparative study of the bivariate, multivariate and machine-learning-based statistical models for landslide susceptibility mapping in a seismic-prone region in China
  3. Earthquake risk assessment using an integrated Fuzzy Analytic Hierarchy Process with Artificial Neural Networks based on GIS: A case study of Sanandaj in Iran

I think proper discussion would highly enhance the level of current study. At this time, it is more like a homework rather than a research.

Author Response

Response to Reviewer -2 Comments

Comments and Suggestions for Authors

I have completed the reviewing of the manuscript “GIS and remote sensing-based multi-criteria analysis for delineation of groundwater potential zones: a case study for industrial zones in Bangladesh”. I have two questions for the authors as follows:

  • What rules are the predicting factors selected based on? The author should give some explanations?
  • What are the relations between the selected factors?

Author Response: Thank you for your comments. Your comments helped us to imprve the quality of our manuscript. The selection of predictors and their inter-relation is important. We added the following texts in the revised manuscript to justify the selection of the predicting factors. Their inter-relations are also provided in the last two sentences.

“The groundwater potential of a region depends on surface hydrological and sub-surface geological conditions. Higher rainfall and abundant surface water bodies enhance groundwater recharge potential. The topography (TWI, plane curvature, profile curvature, etc.) determines water accumulation on the land surface, while soil (type and depth) determines its percolation to the subsurface. Vegetated land and lineaments also help infiltrate surface water to subsurface more. Geology plays a major role in accumulating and transmitting sub-surface water. Therefore, these factors were considered to determine GWPZ in the study area. Generally, a flat terrain covered by favourable LULC and soil with favourable underlying geology has higher groundwater potential. The potential increases when rainfall in the area is more and drainage density is less.” Line 129-139 (Page 4)
